# Picosecond Laser Shock Micro-Forming of Stainless Steel: Influence of High-Repetition Pulses on Thermal Effects

**DOI:** 10.3390/ma15124226

**Published:** 2022-06-15

**Authors:** José Manuel López, David Munoz-Martin, Juan José Moreno-Labella, Miguel Panizo-Laiz, Gilberto Gomez-Rosas, Carlos Molpeceres, Miguel Morales

**Affiliations:** 1Centro Láser, Universidad Politécnica de Madrid, Alan Turing 1, 28038 Madrid, Spain; david.munoz@upm.es (D.M.-M.); juanjose.moreno.labella@upm.es (J.J.M.-L.); miguel.panizo.laiz@upm.es (M.P.-L.); carlos.molpeceres@upm.es (C.M.); miguel.morales@upm.es (M.M.); 2Escuela Técnica Superior de Ingenieros Industriales, José Gutiérrez Abascal 2, 28006 Madrid, Spain; 3Escuela Técnica Superior de Ingeniería y Diseño Industrial, Ronda de Valencia 3, 28012 Madrid, Spain; 4Departamento de Física, Centro Universitario de Ciencias Exactas e Ingeniería Olímpica, Guadalajara 44430, Mexico; gilberto.grosas@academicos.udg.mx

**Keywords:** laser peen forming, simulation, residual stress, LSP, radius of curvature, repetition rate

## Abstract

A study of the peen forming of thin stainless steel metal foils (50 μm thick) using a solid-state ps-pulsed laser, emitting at a wavelength of 1064 nm was conducted. The pitch distance between consecutive laser pulses was kept constant by tuning the laser repetition rate from 0.4 to 10 kHz, and subsequently the scanning speed. The induced bending angle and the radius of curvature were used to measure the effect of the treatment. Their dependence on the pulse energy, the treated area, the distance between lines, and the laser repetition rate was studied. High repetition rates do not allow the sample to cool down, affecting the bending to the point of being negligible. An FEM simulation and experiments were carried out to prove that the increase in temperature due to high repetition rate can relax the stresses induced by laser peen treatment, thus preventing bending in the sample.

## 1. Introduction

Shot peening is widely used to enhance the properties of tool bodies by introducing near-surface compressive residual stresses, which also improves fatigue resistance, crack initiation, and propagation [1]. This is very helpful in materials that are used in disciplines like medicine or the aerospace industry; an example of this would be 316 stainless steel. Laser shock peening (LSP) uses nanosecond (ns) lasers to induce ablation on the materials after generating an expanding plasma followed by a shock wave [2,3]. LSP has been proposed as a competitive alternative technology to classical treatments for the precisely controlled treatment of localized critical areas such as holes, to improve different properties of metals, for example fatigue, corrosion, and wear resistance [4]. It is the shock wave that induces compression stress. However, this is a very inefficient process when there is no confining layer, and the pressure is too low. With the aim of increasing the pressure generated by the plasma, a transparent confining layer is employed. It allows the beam to pass through it with hardly any loss of irradiance, notably increasing the pressure [5,6].

It was quickly observed that LSP could be used to modify the geometry of the treated materials using the plastic changes (due to compressive stresses on the surface) produced by the shock waves [7]. This technique is commonly referred to as laser peen forming (LPF). Some of the advantages of these two laser techniques are their high flexibility (they can be performed in many different setups and patterns), and the fact that they are free of wear and deflection [8]. Ideally, LSP will be a purely mechanical process [7]. LPF along with laser cutting has been used to shape metals like titanium [9]. Simulations using the finite element method (FEM) show that the induced stresses elongate the material in its lateral directions [10,11]. Scanning the laser along the surface, it is possible to microform the sample [12].

Sample thickness has been demonstrated to be an important characteristic, along with pulse energy. If the plastic layer generated by the shock wave is relatively small in depth and length, there will be a rapid stress gradient in which the sample surface suffers maximum compression. This induces a bending moment in the beam direction (positive bending angle or convex curvature). However, if the deformation reaches the other side of the sample thickness, the plastic deformation will rebound so that the bending moment is applied in the opposite direction (negative bending angle or concave curvature) [8].

LSP can be considered to be a cold process, since it is primarily mechanical in nature. The main idea of LSP is to use the shock wave generated by the plasma but, in order to do so, it is necessary to ablate the material, which will change the surface temperature. When a standard high-pulse-energy ns laser is used, the repetition rate will be around the tens of hertz, which allows the sample to cool down between pulses. Ultrashort lasers—pico-(ps), and femtosecond (fs) lasers—exhibit other advantages such as finer and more uniform microstructure and improved microhardness [13]. Such lasers usually work at higher repetition rates (on the order of kHz to MHz), and therefore it is no longer obvious that the process is cold.

A practical industrial example of this technique is the manufacture of suspended magnetic heads for hard disks [14]. It can also be used as an alternative and easier technique for characterizing treatments, instead of measuring superficial stresses; the same bending implies the same treatment, in a similar way to in shot peening [15].

The current paper studies the bending induced in thin 316 stainless steel samples and how this bending depends on the energy per pulse, the treated area, the distance between lines, and the repetition rate. Two magnitudes are studied: the bending angle and the radius of the curvature. Throughout the study, it is intended to demonstrate that the material can be bent accurately, inducing surface tensions as a function of the values of the parameters previously listed. The study also shows that the bending effect is local, and it cannot be scaled by increasing the repetition rate, because the increase in temperature relaxes the superficial stresses previously induced.

## 2. Materials and Methods

This research uses a diode-pumped solid-state ps-pulsed laser (Ekspla Atlantic 355-60), working at 1064 nm with a 13 ps pulse width. Stainless steel metal foils were irradiated to be laser peen formed. The laser repetition rate $\left(f\right)$ was varied from 1 to 20 kHz. The laser pulse energy $\left(E\right)$ used was 115 μJ in most of the experiments, and it was measured using a thermal sensor with an accuracy of 3% placed after the focusing lens.

Figure 1 shows an underwater sample set on top of a holder inside a container. The beam is focused on the cantilever section using a fixed lens (Linos Focus-Ronar) with a focal length of 58 mm. The laser Gaussian beam waist at focus had a radius $\omega_{0}=10\;\mu m$; therefore, the Gaussian peak irradiance was $5.29\frac{\mathrm{TW}}{\;{\mathrm{cm}}^{2}}$ and the Gaussian peak fluence was $73.20\;\frac{J}{{\mathrm{cm}}^{2}}$. The container is part of a closed circuit in which distilled and filtered water (to avoid the slag from the process modifying the experiment) circulates through a system of pipes and a pump.

First, 50 μm-thick cold rolled foils of 316 stainless steel samples were laser cut into a “T” shape (as can be seen in Figure 1 and Figure 2). The top section of the “T” was attached to the edge of a microscope glass slide, leaving the narrower part in a cantilever configuration. That cantilever section was 1 mm wide and 5 mm long. The top side of the sample was irradiated by scanning along the *X*-axis (normal to the longest part of the T), starting and ending at least 1 mm outside of the sample to avoid any effect that acceleration may have on the edges.

A working experiment sketch is shown in Figure 2a. In it, the beam was scanned along the surface by moving the sample with two computer-controlled X and Y linear translation stages. It is worth noting that it begins at the furthest side of the sample; if this were carried out in the other direction, every line would be pushed more and more out of focus. The horizontal pitch $\left(P_{x}\right)$ is the distance between consecutive laser pulses, and will remain 1 μm throughout this study. The vertical pitch $\left(P_{y}\right)$ is the distance between consecutive lines; this latter was varied in different experiments. Throughout this work, the repetition rate will be modified, but it is important to keep in mind that there will always be the same number of pulses per point; for this, the scanning speed $\left(v\right)$ has to be adjusted accordingly. Figure 2b shows a treated sample. The bending angle was measured using bright field microscopy. A blue diode lights a diffusive screen to scatter light that will be collected by a 4× microscope objective.

The diameter of the zone affected by a focused single-laser pulse measured by confocal microscopy was 48 μm, around two and a half times the beam waist $\left(2\omega_{0}=20\;\mu m\right).$ Therefore, the beam overlapping using $P_{x}=1\;\mu m$ was 97.9%, which will remain constant throughout all of the experiments, no matter what else is changed.

## 3. Results

The length $\left(L\right)$ of the treated area is related to the bending angle $\left(\delta \right)$ through the radius of curvature $\left(R\right)$.
(1)L=Rδ

Figure 3 shows an example of an irradiated sample. The sample was completely horizontal before the laser treatment, and after it, the sample was folded, with a bending angle δ±Δδ. The relative uncertainty of the bending angle ranges between 1% and 2%.

The non-laser-treated regions remain straight, and the deformation is limited to the irradiated area. Therefore, the slopes of the non-treated zones can be compared, and the angle δ can be calculated as a certain difference between those slopes. This allows us to obtain R.
(2)$$R=\frac{L}{\delta }=\frac{L}{\pi \pm \alpha_{1}-\alpha_{2}}$$

A set of experiments was carried out to study the effect of pulse energy E using L values of 1 mm and a 20 μm vertical pitch, with E ranging from 35 to 111 μJ, and the speed and repetition rate remaining constant, with values of 1 kHz and $1\;\frac{\mathrm{mm}}{s}$, respectively; therefore, $P_{x}$ is 1 μm. On the basis of the results shown in Figure 4, the low pulse energy values barely induce bending, but with increasing energy, the microforming effect becomes more remarkable, producing larger bending angels δ, while at the same time, R decreases. Since we are interested in the effects of larger δ (because they are easier to measure), this experiment justifies us working at maximum E.

Although, in principle, δ and R are equivalent descriptions, the information on the radius of curvature is more useful. Figure 5 shows a set of experiments studying the effect of varying the size of the treated area using a pulse energy of 115 μJ and $P_{y}\;20\;\mu m$, while the speed and repetition rate are again 1 kHz and $1\frac{\mathrm{mm}}{s}$, respectively, and the horizontal pitch $P_{x}$ is 1 μm. With increasing treated area, δ increases proportionally, whereas R remains constant.

In the previous analysis, the length of the treated area was varied, but $P_{y}$ was kept constant. The subsequent one was conducted using high energy per pulse (111 μJ), a 1 kHz repetition rate (and therefore a $1\;\frac{\mathrm{mm}}{s}$ scanning speed), and the same $P_{x}$ of 1 μm, while varying the vertical pitch among 10, 20, 40, and 60 μm, resulting in treated areas with lengths of 0.5, 1, 2 and 3 mm, respectively (every sample has the same number of lines). The bending angle and the radius of curvature are plotted in Figure 6, there is no meaningful variation in the angle with changing $P_{y}$; however, there is a linear relation between $P_{y}$ and R.

According to Stuart et al., ps and fs lasers should be capable of performing ablation without increasing (or at least doing so only slightly) the temperature of the sample surface [16]. However, when using high-repetition-rate ps lasers for LPF, that slight temperature increase per pulse can accumulate, affecting the performance of the treatment. Figure 7 shows the effect of the repetition rate. All of the samples were shot with 115 μJ in an area with a length of 1 mm using a 20 μm vertical pitch. To preserve the number of pulses per spot, the scanning speed was tuned accordingly, always bearing in mind that the density of the energy deposited onto the sample will be the same every time. Those samples irradiated at 1 kHz or lower showed large bending angles. R seems to grow exponentially with increasing f, which implies that those laser treatments performed at a higher repetition rate are less effective for bending the sample. Moreover, at the highest laser repetition rate used, the bending angle tended to zero.

To understand the effect of f, FEM simulations (see Appendix A) and a new set of experiments were conducted. Figure 8 plots the effect of different repetition rates computed on the basis of FEM thermal simulations of steel temperature using ps radiation. In summary, a point of the surface receives the energy of a Gaussian beam, and the beam shifts with every pulse; this means that the energy received by the point is different with every pulse. This increases when the beam reaches the point, and decreases when it moves away. As f increases, the material heats, and its temperature at the moment at which it receives the next pulse will be higher.

Figure 9 shows the envelope of the minimum values of temperature before every pulse; this time, a wider range of frequencies is shown. The manner in which the temperature piles up can be clearly seen. Additionally, the inset plots the maximum cooling down temperature depending on the repetition rate. For the range of frequencies employed in this study, this relation remains linear.

Figure 10 presents a comparison of the importance of pulse energy and repetition rate. All samples were treated on an area with a length of 1 mm and, as in the rest of this research, the scanning speed was tuned according to the repetition rate to keep the number of pulses per point constant. Figure 10a shows the case of the lowest repetition rate $\left(1\;\mathrm{kHz}\right)$ and high energy $\left(111\;\mu J\right)$, the result is a sample with a large bending angle. Figure 10b also presents the case in which a low repetition rate was employed, but the energy was minimal (34 μJ), and it can be seen tht, while the sample bends, the angle is so small that it is difficult to measure. Figure 10c,d were obtained after working at the highest repetition rate $\left(20\;\mathrm{kHz}\right)$, with the first one employing 111 μJ per pulse and the former 34 μJ. No measurable bending angles were obtained.

## 4. Discussion

The results presented in Figure 5 and Figure 6 prove that the process is differential, which means that the effects of the lines are additive. In both cases, each line is bent by the same quantity: a differential angle dδ. In the experiment presented in Figure 5, with the same separation between the lines $P_{y}$, each line in all samples contributes with the same radius of curvature, and therefore, the total angle is the sum of all of the differential angles, and the radius of curvature remains constant. On the other hand, in the experiment presented in Figure 6, the separation between the lines was modified in every sample and, although the differential angle remains the same, the radius of curvature due to the lines now changes from sample to sample. If the deformation produced by a single line is known, the effect of k lines will be k times the bending angle (but the radius of curvature will be unaffected by this). So far, it has been shown that the induced compressive stresses can be controlled, and that bending is a tool available for checking their behavior. Now, the goal is to scale the process, so the speed of the treatment will be increased by using higher repetition rates (with scan speeds adjusted accordingly).

Traditionally in LSP processes, the repetition rate is so low (on the order of the hertz or tens of hertz) that the sample is always able to cool down completely between pulses. Therefore, the repetition rate has no impact on the process. However, the experiments conducted here were performed using repetition rates that were higher by orders of magnitude. Figure 7 showed that when the repetition rate is very high, the samples exhibit a higher radius (they barely bend). Our hypothesis is that high repetition rates do not allow the sample to cool down completely, and due to this fact, the following pulses increase the temperature. The sample does not heat up enough to melt but to relax the stresses that the shock waves had induced. In other words, the mechanical component of the process is neutralized by the thermal one. This result is in good accordance with previous works [17,18].

In the simulations presented in Figure 8, it can be seen that 1 kHz (low f) allows the material to cool down, and every pulse meets a sample at room temperature. If the repetition rate increases enough (like the 5 and 15 kHz examples) the next pulse will occur before the sample has cooled down completely, leading to material heating. Figure 9 shows that for a wider range of f, the simulation predicts a linear relation between the maximum cool-down temperature of the surface and the repetition rate. The thermal relaxation of the residual stresses induced by LSP has been reported previously [19,20] in annealing processes with temperatures in the range of 200–650 °C, although this effect is more important for longer times. In this case, the annealing time is short, but the affected volume is also smaller than in previous studies.

Figure 10 is the experimental confirmation of the idea introduced by the simulation results. When the repetition rate is low (1 kHz) the difference in energy (a. 111 μJ and b. 34 μJ) has a significant impact on the sample, because the mechanical effect of the shock wave is the main process, and it depends strongly on the pulse energy, with every shot inducing compressive stress that remains in the material. Nevertheless, high repetition rates (20 kHz) and high energies per pulse (c. 111 μJ) neutralize and relax the tensions induced by the shock wave, and thus, there is no visible difference between this and a sample treated using minimum energy (d. 34 μJ).

It has been shown that relaxation is affected by the residual stress state itself and by the material state, such as the material type and its microstructure [21]. However, in a previous study, performed with the same laser source and similar materials [22], no effect in the microstructure was observed. The laser pulse duration was so short that matter diffusion processes did not occur; furthermore, with respect to temperature increase, even if it occurred, it was irrelevant to the production of microstructural modifications.

Regarding the mechanical properties of the samples, hardness tests were carried out using a Shimadzu DUH-211S dynamic ultra-micro hardness tester. The T-shaped parts were too thin and deformed for the analysis to be performed, so the same process was carried out on thicker stainless steel 316L samples (2 mm), in order to prevent them from bending because of the laser interaction. On these new thicker parts, there was no improvement in hardness compared to the non-treated sample. The value in Vickers of the substrate was (294 ± 22) HV, whereas the samples were treated using 111 µJ energy pulses, and P_x_ of 1 µm and 20 µm separation between lines; one sample was scanned at 20 $\frac{\mathrm{mm}}{s}$ (20 kHz repetition rate) and the other at 1 $\frac{\mathrm{mm}}{s}$ (1 kHz repetition rate), and their microhardness values were (253 ± 73) HV and (233 ± 45), respectively. In the case of austenitic steels, the long maintenance (in the range of minutes) at medium temperatures forces chrome carbides to segregate in the grain boundaries (leading to a sensitized structure, more prone to suffering from intergranular corrosion). The grain size of a polycrystalline material increases over time with increasing temperature, but only for long periods of maintenance. Laser Shot Processing involves the very brief input of heat into the material. Even though the laser pulse repetition rate is high, the total thermal load on the surface of the part is not high enough to induce microstructural changes. Therefore, the effects of both carbide precipitation and grain growth can be completely neglected in LSP processes.

## 5. Conclusions

The parameters that were frequently used were energy pulse between 111 and 115 μJ, frequency of 1 kHz $\left(v=1\frac{\mathrm{mm}}{s}\right)$, $P_{x}$ of 1 μm, and $P_{y}$ of 20 μm.

The effect of E shows that the more energetic our process is, the greater the bending. The bending angle can be precisely adjusted using specific conditions of $E,\;f,\;v,{\;P}_{x}$ and $P_{y}$. The largest bending angle was achieved, this being 25.3°, was achieved with the maximum energy per pulse $\left(111-115\;\mu J\right)$.

Varying L but keeping $P_{y}$ constant varies the bending angle in a linear fashion but the radius of curvature remains constant at around 1.10±0.11 mm. The effect of the $P_{y}$ (varying L accordingly and maintaining the same number of lines) barely changes the δ of the sample $\left(24.4\pm 1.9\right){}^{\circ}$. As a consequence, R increases with increasing length of the treated area. These two experiments confirm that the process is local, since every treated line similarly curves the sample.

LSP with ps lasers usually assumes that the pulse time is short enough to avoid thermal effects, and thus the effect of the repetition rate has been neglected, but if that were the reason, experiments using the same E,L, and number of pulses per position should be unaffected by changes in f, but when we try to scale the process using high repetition rates (above 1 kHz), a stacking effect in temperature appears. The energy is not completely dissipated, locally raising the temperature of the sample and relaxing the stresses that the shock waves had originally created.

## Figures and Tables

**Figure 1 materials-15-04226-f001:**
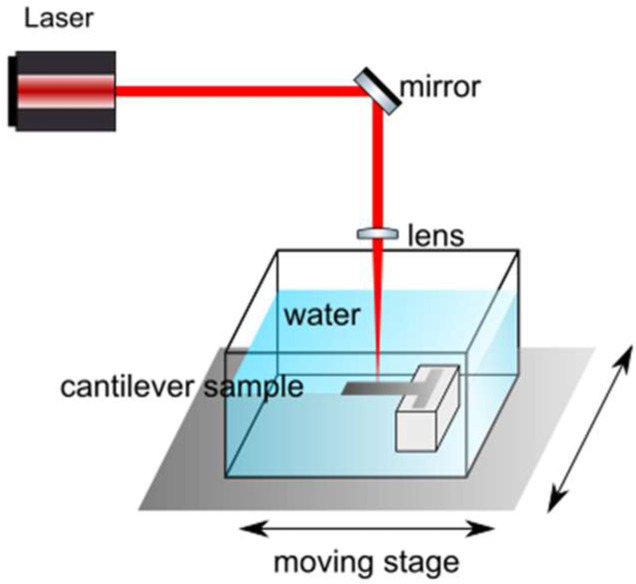
Experimental setup and laser scanning. A moving stage moves the container in the X and Y directions, so the focused Gaussian beam works on the cantilever section of the sample, which is submerged in water (confining medium).

**Figure 2 materials-15-04226-f002:**
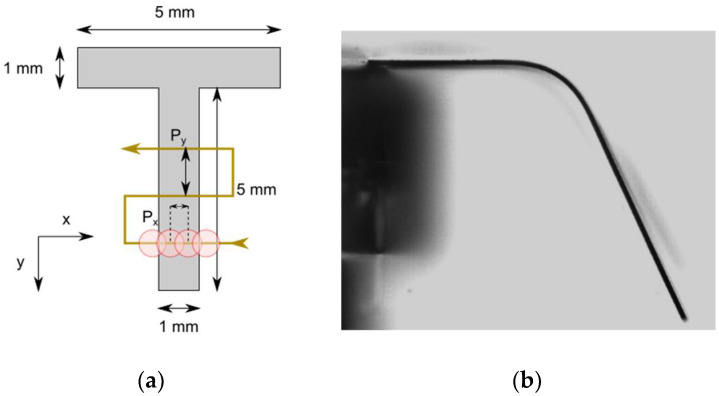
(**a**) The laser begins working at the furthest position of the cantilever section outside the sample. The stage moves as the axes are displayed; in this way, the working area is always in focus. The picture also shows $P_{x}$ and $P_{y}$. (**b**) A 316 steel sample after the process.

**Figure 3 materials-15-04226-f003:**
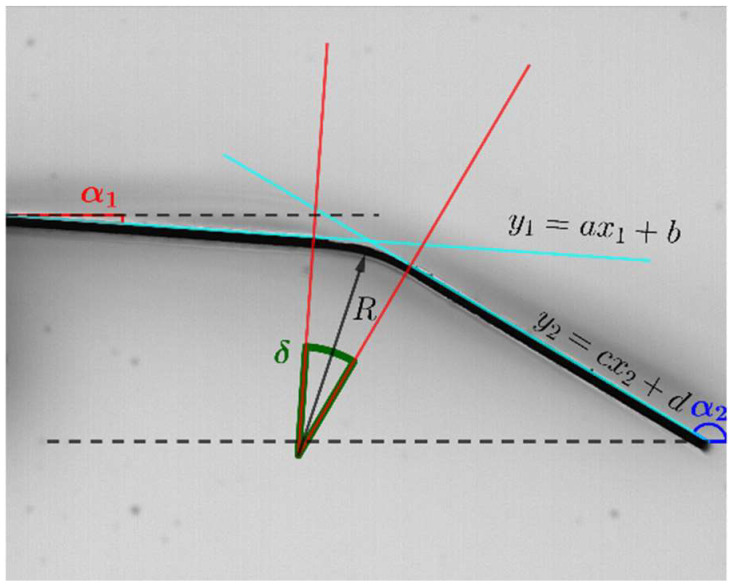
$\alpha_{1}$ and $\alpha_{2}$ can be obtained through the slopes of the straight lines, and can be used to compute both δ and R.

**Figure 4 materials-15-04226-f004:**
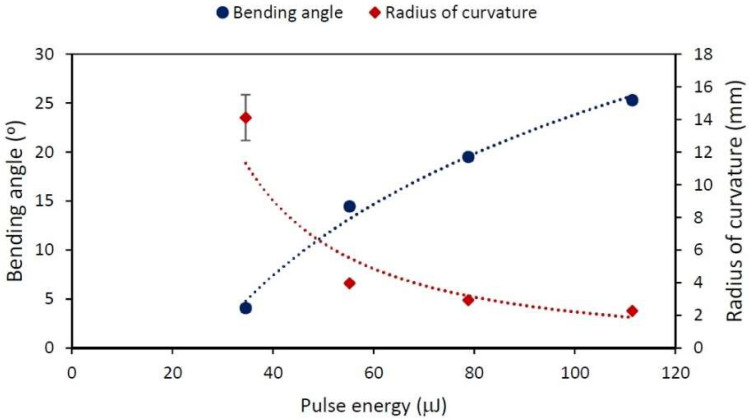
Bending angle (round points) and radius of curvature (diamonds) dependence on pulse energy ranging from 35 to 111 μJ. Treated area of 1 mm length, $P_{y}=20\;\mu m$, $P_{x}=1\;\mu m$, f=1 kHz and $v=1\frac{\mathrm{mm}}{s}$. The lines are visual guides.

**Figure 5 materials-15-04226-f005:**
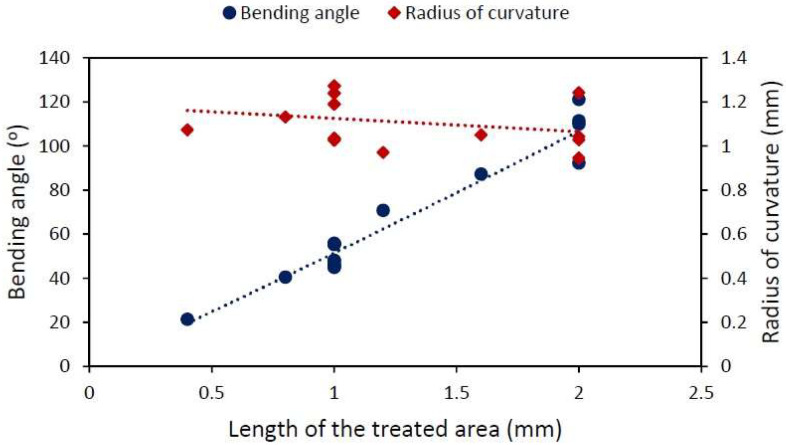
Bending angle (round points) and radius of curvature (diamonds) dependence on the length of the treated area ranging from 0.4 to 2 mm. E=115 μJ, $P_{y}=20\;\mu m$, $P_{x}=1\;\mu m$, f=1 kHz and $v=1\frac{\mathrm{mm}}{s}$. The lines are visual guides.

**Figure 6 materials-15-04226-f006:**
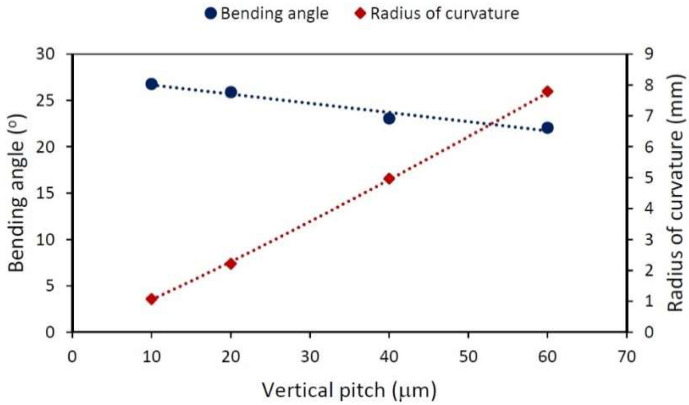
Bending angle (round points) and radius of curvature (diamonds) dependence on vertical pitch ranging from 10 to 60 μm. The length of the treated area ranged accordingly from 0.5 to 3 mm, while E=111 μJ, $P_{x}=1\;\mu m$, f=1 kHz and $v=1\frac{\mathrm{mm}}{s}$. The lines are visual guides.

**Figure 7 materials-15-04226-f007:**
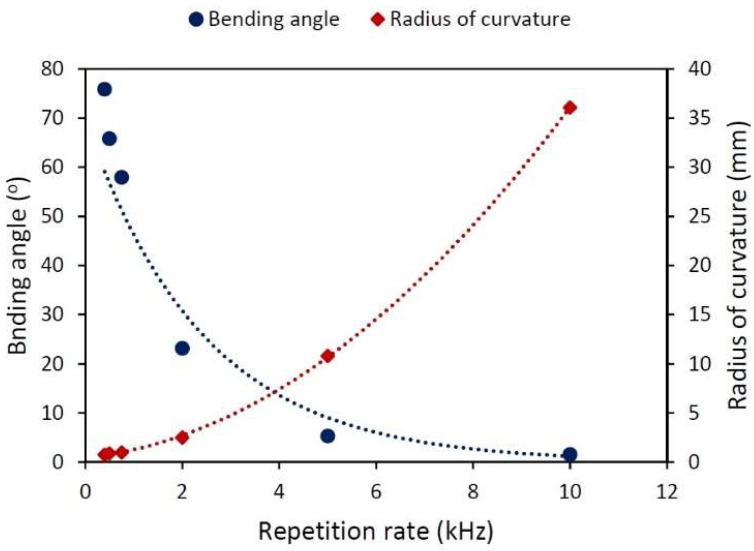
Bending angle (round points) and radius of curvature (diamonds) dependence on repetition rate ranging from 0.4 to 10 kHz. Scanning speed ranging accordingly from 0.4 to $10\;\frac{\mathrm{mm}}{s}$. Treated area with a length of 1 mm, $P_{y}=20\;\mu m$, $P_{x}=1\;\mu m$, and E=115 μJ. The lines are visual guides.

**Figure 8 materials-15-04226-f008:**
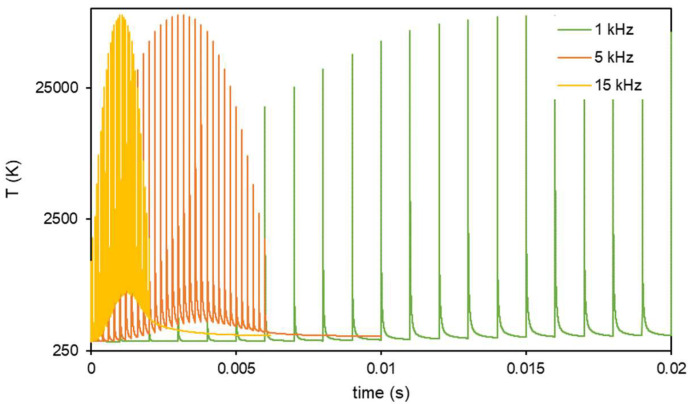
Steel temperature when irradiated with ps-laser pulses at different repetition rates (FEM). Only 1, 5, and 15 kHz (green, orange, and yellow series, respectively) are shown, for clarity. Cool-down temperature becomes higher with increasing repetition rate.

**Figure 9 materials-15-04226-f009:**
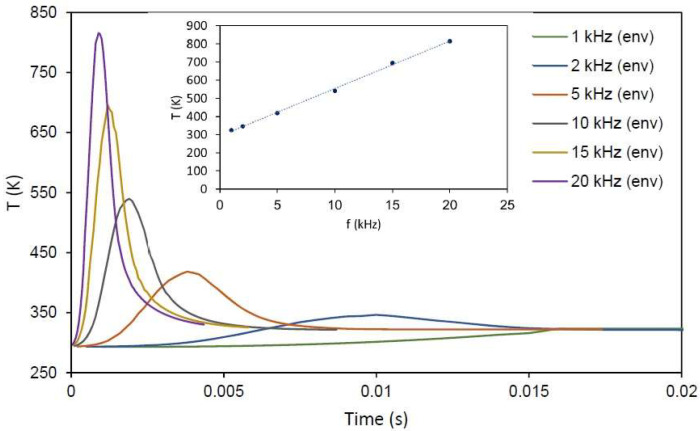
Envelope of the cool-down temperature depending on the repetition rate. Inset shows the linear behavior of the maximum cool-down temperature with frequency. Lines are visual guides.

**Figure 10 materials-15-04226-f010:**
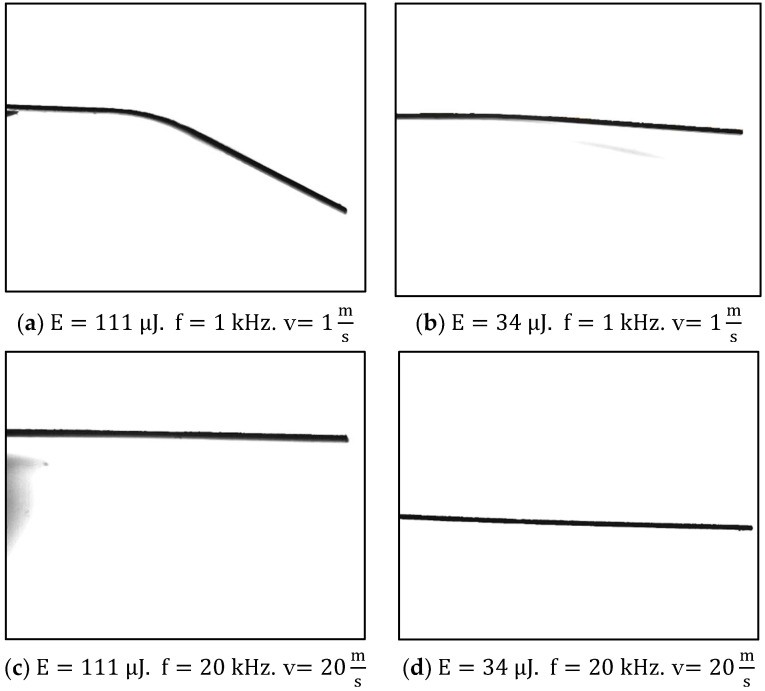
Low repetition rate $\left(1\;\mathrm{kHz}\right)$ and different energies ((**a**) 111 μJ and (**b**) 34 μJ ) compared to high repetition rate $\left(20\;\mathrm{kHz}\right)$ with the same energies ((**c**) 111 μJ and (**d**) 34 μJ ). Configuration a. induces bending while the others, either by using little energy or by using a high repetition rate, do not produce measurable bending.

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
