# Peer review of "Picosecond Laser Shock Micro-Forming of Stainless Steel: Influence of High-Repetition Pulses on Thermal Effects"

_materials, 2022, doi:10.3390/ma15124226_

Round 1

Reviewer 1 Report

The study lacks an aspect related to the analysis of structural changes taking place during the process. The analyzes show that the temperatures are high. Will it not have an effect on grain size, supersaturation processes in stainless steel? Why did the authors not perform at least microhardness measurements to determine changes in material properties in the shaping area? The presented technique is very interesting and the obtained results indicate that in thin layers it is possible to shape elements with this technique. In one study, the problem of energy consumption was completely ignored. How does it compare with classical plastic working? In my opinion, the work has great potential, but the lack of any material analyzes is glaring. For example, in line 225 the authors write that the microstructure has been investigated. However, no own research in this regard is available. The authors cite the publication [15], however, they do not comment on its results in any way. It should be remembered that the interaction with energy always affects the microstructure of the material and, consequently, its properties.

Author Response

We thank the referee for its constructive criticism, and time spent to analyze this manuscript, The reported issues along with their responses, and explanations are listed below:

  1. As a response of the referee comment aspect related to the analysis of structural changes taking place during the process, we have added three paragraphs in section 4. Discussion (lines 259-279). Presenting results from previous works and from currents experiments stating there are no changes in microhardness mostly because the grain size of a polycrystalline material grows over time the higher the temperature is, but only for long maintenance. In order to check that microhardness measurements were carried out. 50 micron thick samples cannot be measure so we changed to a thicker sample (in the order of the mm). In that case, values in the range of 250-270 HV were obtained.

Substrate: (294 ± 22) HV

20 kHz: (253 ± 73) HV

1 kHz: (233 ± 45) HV

  1. In the case of the treated samples, the roughness affects the microhardness measurement, making it less precise, but no significant increase is observed.

  1. As a response of the referee comment aspect related to energy consumption, we want to clarify that in this process interaction time is restricted to plasma duration (in ns pulses around 0.05 us, in ps regime around 0.01 us) while in classical plastic working like shot peening interaction time is related to surface/surface contact (around 0.5-1 us). Longer interactions times are usually related to larger affected volumes. In any laser process energy efficiency is lost in order to achieve a higher quality energy (short time and small area that provide high intensity). In this case shorter interaction times and small area allow us to control bending in a very small thickness sample that would be complicated with a classical approach.

  1. As a response of the referee comment aspect related to the lack of research on the microstructure we added a previous work (references 21) in section 4. Discussion.

  1. As a response of the referee comment aspect related to the change in microstructure we have performed measurements on several samples with no meaningful changes in microstructure as it is stated in section 4. Discussion (lines 259-279).

Reviewer 2 Report

The Manuscript is very interesting and useful to research and industry professionals, and my opinion is that it can be accepted after major revision.  The authors should estimate the energy density (fluence) for the laser energies applied, and try to discuss the results from that point of view. The fluence is more valuable information than output pulse energy. 

Also, figure 3 has a poor quality, and it should be improved graphically.

Author Response

We thank the referee for its constructive criticism, and time spent to analyze this manuscript, The reported issues along with their responses, and explanations are listed below:

  1. As a response of the referee comment aspect related to the laser fluence, we have added it in section 2. Materials and Methods (line 87) a value of 5.29 TW/cm^2 is now written. This value is three orders of magnitude higher than other laser sources in ablation condition underwater studied in the bibliography like reference 14.

  1. As a response of the referee comment aspect related to figures low quality, now all the figures has been revised and improved.

Reviewer 3 Report

The article is devoted to the development and research of new methods of material processing. Laser technologies are proving to be efficient and promising for new application areas. The article may be useful to readers. However, while studying the article, I had questions and comments to the authors.

  1. In the Annotation you used the term "sheet". When the thickness of rolled products is 50 microns, it is more correct, I think, to write "foil".

2. The introduction of the article does not indicate the scope of parts where bending to the required angle is required with a foil thickness of 50 microns. Why is it necessary to bend such thin products at a fixed angle?

3. The article does not say what equipment or tools were used to fix the obtained bending angle. Was it a microscope?

4. In the description of the test material, only the brand is indicated. In what state (after cold rolling, heat treatment) was the studied foil? What are the mechanical properties of steel?

5. It would be useful to show how laser processing has affected the microstructure of steel. What changes did she make? How will this affect the performance properties of the steel and the part?

Author Response

We thank the referee for its constructive criticism, and time spent to analyze this manuscript, The reported issues along with their responses, and explanations are listed below:

  1. As a response of the referee comment aspect related to the terms “sheet” and “foil”, the suggestion has been taken in consideration and the document has been changed accordantly.

  1. As a response of the referee comment aspect related to why is it necessary to bend such thin products at a fixed angle, we have added in section 1. Introduction (lines 73-76) a paragraph naming the manufacturing of suspended magnetic heads for hard disks and also that this process is interesting as an easier characterization technique of measuring superficial stresses.

  1. As a response of the referee comment aspect related to the lack of information of the equipment used to obtained the bending angle, we have added lines 111-113 in section 2. Materials and methods explaining how a blue diode light is used to apply bright field microscopy by scattering light on a diffusive screen. A 4x microscope objective was used.

  1. As a response of the referee comment aspect related to the state of the test material, we added in line 95, section 2. Materials and methods that the studied samples were -thick cold rolled foils of 316 Stainless steel. The referee also asked about the mechanical properties of steel, since it is a 316 its minimum Yield Strength is 206.8 MPa and its minimum Tensile Strength is 517.1 MPa.

  1. As a response of the referee comment aspect related to the effect on the microstructure, we added reference 21 in section 4. Discussion were we state that there was no effect in the microstructure since it is a fast action and matter diffusion processes do not happen.

Round 2

Reviewer 1 Report

Thank you for including your comments. I wish you further success.

Reviewer 2 Report

The authors revised the Manuscript, and in my opinion it is now acceptable for publishing. 

Reviewer 3 Report

The authors of the article have made changes to the paper according to my comments. I recommend accepting the article in its present form for publication.